# Hereditary Diffuse Gastric Cancer: A Comparative Cohort Study According to Pathogenic Variant Status

**DOI:** 10.3390/cancers12123726

**Published:** 2020-12-11

**Authors:** Tim Marwitz, Robert Hüneburg, Isabel Spier, Jan-Frederic Lau, Glen Kristiansen, Philipp Lingohr, Jörg C. Kalff, Stefan Aretz, Jacob Nattermann, Christian P. Strassburg

**Affiliations:** 1Department of Internal Medicine I, University Hospital Bonn, 53127 Bonn, Germany; tim.marwitz@ukbonn.de (T.M.); jacob.nattermann@ukbonn.de (J.N.); Christian.Strassburg@ukbonn.de (C.P.S.); 2National Center for Hereditary Tumor Syndromes, University Hospital Bonn, 53127 Bonn, Germany; isabel.spier@uni-bonn.de (I.S.); Jan-Frederic.Lau@ukbonn.de (J.-F.L.); glen.kristiansen@ukbonn.de (G.K.); Philipp.lingohr@ukbonn.de (P.L.); kalff@uni-bonn.de (J.C.K.); Stefan.Aretz@uni-bonn.de (S.A.); 3Institute of Human Genetics, University Hospital Bonn, 53127 Bonn, Germany; 4Institute of Pathology, University Hospital Bonn, 53127 Bonn, Germany; 5Department of Surgery, University Hospital Bonn, 53127 Bonn, Germany

**Keywords:** hereditary cancer, gastric cancer, *CDH1*, *CTNNA1*, HDGC

## Abstract

**Simple Summary:**

Pathogenic germline variants in *CDH1* and *CTNNA1* lead to a high prevalence of diffuse gastric cancer and breast cancer in carrier families. However, in most families with hereditary diffuse gastric cancer syndrome, the underlying genetic alteration remains elusive. In this cohort study we report on patients who met the criteria for genetic testing and compare carriers of a pathogenic *CDH1/CTNNA1* variant with patients without diagnosis of a pathogenic germline variant. Gastric cancer prevalence in both groups was high and patients were diagnosed at an early age. Prevalence of breast cancer in female patients with a confirmed pathogenic variant (*CDH1*) was lower than previously described. Given the early age of onset in families with a suspected hereditary etiology of gastric cancer, further research is required aimed at improving clinical management and potentially the outcome in these patients.

**Abstract:**

Hereditary diffuse gastric cancer (HDGC) is an inherited cancer susceptibility syndrome characterized by an elevated risk for diffuse gastric cancer (DGC) and lobular breast cancer (LBC). Some patients fulfilling the clinical testing criteria harbor a pathogenic *CDH1* or *CTNNA1* germline variant. However, the underlying mechanism for around 80% of the patients with a family or personal history of DGC and LBC has so far not been elucidated. In this cohort study, patients meeting the 2015 HDGC clinical testing criteria were included, and subsequently, *CDH1* sequencing was performed. Of the 207 patients (161 families) in this study, we detected 21 pathogenic or likely pathogenic *CDH1* variants (PV) in 60 patients (28 families) and one *CTNNA1* PV in two patients from one family. Sixty-eight percent (*n* = 141) of patients were female. The overall PV detection rate was 18% (29/161 families). Criterion 1 and 3 of the 2015 HDGC testing criteria yielded the highest detection rate of *CDH1/CTNNA1* PVs (21% and 28%). PV carriers and patients without proven PV were compared. Risk of gastric cancer (GC) (38/62 61% vs. 102/140 73%) and age at diagnosis (40 ± 13 years vs. 44 ± 12 years) were similar between the two groups. However, GC was more advanced in gastrectomy specimens of patients without PV (81% vs. 26%). LBC prevalence in female carriers of a PV was 20% (*n* = 8/40). Clinical phenotypes differed strongly between families with the same PV. Emphasis should be on detecting more causative genes predisposing for HDGC and improve the management of patients without a proven pathogenic germline variant.

## 1. Introduction

Despite an overall declining incidence, gastric cancer (GC) still represents one of the leading causes of cancer-related death worldwide [1]. In 10% of GC cases family clustering is observed, which points towards a potentially hereditary disease, although the underlying genetic disorder remains unclear in the majority of cases [2].

Hereditary diffuse gastric cancer (HDGC) is clinically defined by an increased occurrence of diffuse gastric cancer (DGC) and lobular breast cancer (LBC). In 1998 pathogenic germline variants (PV) in the *CDH1* gene responsible for multiple, early-onset diffuse gastric cancer in three Maori kindreds were identified [3].

In 1999 the International Gastric Cancer Linkage Consortium (IGCLC) first published guidelines to select patients eligible for screening of pathogenic *CDH1* germline variants [4]. Since then, testing criteria have been revised multiple times [5,6]. In August 2020 the IGCLC published updated guidelines broadening the testing criteria (Table 1) [7].

*CDH1*, a cancer susceptibility gene, encodes for E-Cadherin, a cell adhesion protein regulating epithelial architecture and cellular polarity [8]. HDGC is an autosomal dominant disorder, occurring with an estimated incidence of 5–10 per 100,000 births [3,7]. In patients with heterozygous *CDH1* PVs, gene inactivation of the second allele by somatic alterations leads to the development of DGC and LBC, whereas frequencies of other cancer types seem not to be affected [7,9,10]. Cumulative risk for developing DGC in PV carriers has been reported to be approximately 70% for men and 56% for women at the age of 80 years [10]. In more recent studies, however, the risk for development of GC in PV carriers not preselected by HDGC testing criteria was found to be markedly lower with 37–42% among men and 24–33% among women [11,12]. In female carriers of a PV in *CDH1* the risk of developing LBC is estimated at 39–55% [7]. Additionally, congenital anomalies such as cleft lip/palate or blepharocheilodontic syndrome are common in carriers of some *CDH1* variants [13,14].

The clinical management of individuals varies depending on the incidence of DGC and LBC in the pedigree. PV carriers with a family history of DGC are advised to undergo prophylactic total gastrectomy (PTG), which often reveals multiple foci of signet ring cell carcinoma (SRCC) without invasive growth [15,16,17]. Although the perfect timing of PTG is still unclear, surgery is generally recommended between 20 and 30 years of age as invasive carcinomas are rare before the age of 20 [6,18].

For patients who refuse or postpone PTG, annual surveillance endoscopies are recommended. However, SRCC is usually small and multifocal in asymptomatic PV carriers, which hampers endoscopic carcinoma detection in *CDH1* carriers and neither endoscopic ultrasound, chromoendoscopy, nor confocal endoscopic microscopy have been shown to increase detection rate of early SRCC [19,20,21,22,23,24].

Following the “Cambridge protocol”, gastric mucosa should be carefully examined with at least 28–30 random biopsies taken from all anatomical regions of the stomach [6].

Carriers of a *CDH1* PV with a personal or family history of LBC but not DGC are categorized as having hereditary lobular breast cancer (HLBC) and should undergo annual endoscopic surveillance [7]. In these patients, PTG can be considered but is recommended only if SRCC is detected during surveillance endoscopy [7]. For female variant carriers, surveillance additionally includes an annual mammogram and/or breast MRI beginning at age 30. Risk-reducing mastectomy can be considered [25].

The term HDGC-like refers to families who meet criteria 1 or 2 of the 2020 testing criteria but without proven PV. For those PTG is not advised and annual surveillance by upper gastrointestinal endoscopy is suggested. Depending on the endoscopic findings, the surveillance interval may be increased after two years [7].

Recently, another genetic variant predisposing for DGC has been identified. In 2013 a loss of Alpha-Catenin expression caused by germline PVs in the *CTNNA1* gene and somatic silencing of the wildtype allele was found in gastric cancer samples of a large HDGC pedigree [26]. Although data on families with germline PVs in *CTNNA1* are scarce, an increased risk for DGC and breast cancer (BC) has been reported, even though the histologic subtype of BC cannot be specified until now [27,28,29].

*CDH1* PVs are only present in 14–19% of patients who fulfill the testing criteria and 65% of carriers of a pathogenic *CDH1* germline variant have been found not to meet the 2015 diagnostic criteria [10,30,31,32]. However, this finding should be interpreted with caution. First, multiplex gene panel testing was performed without consideration of any clinical criteria. Second, to be able to classify a genetic variant, data on its frequency in a large healthy control population is needed. As currently available databases may be limited regarding data quality and reported details on associated phenotypes are clinically relevant, underestimation of variants may occur.

Previous studies focused mainly on patients with a PV. Our study aimed to evaluate the yield of detection of *CDH1* and *CTNNA1* PV according to clinical criteria. Furthermore, we aimed to compare patients with and without PV concerning GC prevalence, age of onset, and other occurring cancers.

## 2. Results

### 2.1. Characteristics of Patient Cohort and Genetic Testing

A total of 242 patients from 177 families were initially enrolled in our study. Twenty patients declined genetic testing or died before testing was conducted and were therefore excluded. Eight relatives of patients with proven germline PV did not meet inclusion criteria (history of DGC or LBC and/or detection of a PV) and therefore were not considered for further analyses. In addition, seven patients were excluded as they met neither the 2015 HDGC testing criteria nor the criteria published in recently updated guidelines. In total, 35 patients were excluded.

In the remaining 207 patients (141 female (68%), 66 male (32%)), belonging to 161 families, sequencing of the *CDH1* gene was performed. In nine families (21 individuals) the *CTNNA1* gene was additionally analyzed in the index patient.

Genetic testing identified 60 individuals in 28 families as carriers of a pathogenic or likely pathogenic *CDH1* germline variant. Five patients in three different families were diagnosed with a variant of uncertain significance in the *CDH1* gene. Figure 1 provides an overview of the study population, as well as the incidence of GC according to the results of genetic testing.

In one female index patient with a personal history of DGC, a PV in the *CTNNA1* gene was detected. Her healthy sister was diagnosed with the same variant.

Average age at inclusion in this study was 47 ± 15 years (range 16–87 years). Overall, prevalence of GC (*n* = 142/207) and DGC (*n* = 138/207) was 69% and 67%, respectively. The average age at GC diagnosis was 43 ± 12 years (range 16–74 years).

Thirteen percent (*n* = 18/141) of female patients included in this study were diagnosed with LBC at a median age of 51 ± 7 years (range 38–65 years). Table 2 shows patient demographics, cancer prevalence, age of onset, and gastrectomy data.

Two patients with other cancer types were identified. One patient without a PV had been previously diagnosed with lung cancer. A patient with a pathogenic *CDH1* variant (c.1679C > G) had a personal history of microsatellite-stable colorectal cancer at the age of 30 years. There were no reports of congenital anomalies such as cleft lip/palate or blepharocheilodontic syndrome in this cohort, with the exception of a 10-year-old child of a mother with a *CDH1* PV (c.1137G > A).

### 2.2. HDGC Criteria

Table 3 shows the number of patients who met the 2015 HDGC testing criteria based on findings in genetic testing.

Overall, 18% (29/161) of index patients were diagnosed with either a *CDH1* or a *CTNNA1* PV. In total, a *CDH1/CTNNA1* germline PV was found in 62 of 207 (30%) patients. In five patients that met the 2020 but not the 2015 testing criteria no *CDH1/CTNNA1* PV was detected. Independent of the PV status, most index patients met criterion 1 (two cases of gastric cancer regardless of age with at least one confirmed DGC) (97/161; 60%) and criterion 2 (one case of DGC before age 40) (87/161; 54%) of the 2015 HDGC testing criteria. Criterion 3 (personal or family history of DGC and LBC with one diagnosed before age 50) was met by 40 of 161 (25%) index patients. Sixty-six index patients (41%) met two or more of the 2015 IGCLC criteria.

Frequency of index patients carrying a pathogenic variant in *CDH1/CTNNA1* was highest in families meeting criterion 3 (11 of 40 (28%)) and criterion 1 (20 of 97 (21%)). A *CDH1/CTNNA1* PV was present in 13 of 87 (15%) index patients fulfilling criterion 2. The PV detection rate was 23% (15/66) in index patients fulfilling more than one criterion of the 2015 testing criteria.

### 2.3. Helicobacter pylori (HP) Infection

A total of 28/207 (14%) patients were histopathologically diagnosed with HP infection in either gastric biopsy or gastrectomy specimen. Twenty-four (86%) of the HP(+) were diagnosed with DGC and one patient with intestinal type gastric cancer. Age of diagnosis of gastric cancer in this subgroup was 49 ± 15 years (range 16–72 years).

Eighteen (13%) patients without a *CDH1/CTNNA1* PV were diagnosed with HP infection, all except one patient (95%) of those were diagnosed with GC at a mean age of 52 ± 13 years (range 34–72 years).

Eight (13%) carriers of a *CDH1* PV had HP infection and six were diagnosed with DGC at a mean age at diagnosis of 42 ± 21 years (range 16–72 years). Three were early SRCCs diagnosed after PTG. The remaining three patients presented with advanced stage GC.

Two patients with a *CDH1* VUS and DGC at age 37 and 45 years were diagnosed with HP.

### 2.4. Patients with CDH1 or CTNNA1 PV

Overall, we detected 21 different pathogenic or likely pathogenic *CDH1* variants in 28 families and one *CTNNA1* PV in one family. Table 4 shows the detected *CDH1* and *CTNNA1* germline variants, the number of family members included in this study (includes only PV carriers and predictively tested relatives who are carriers of the variant), the number of GC and BC cases in the pedigree as well as mean age of onset of manifest cancer. Out of the 62 patients with a PV or likely PV in the *CDH1* or *CTNNA1* genes 22 (35%) were men and 40 (65%) women. Mean age at study inclusion was 42 ± 14 years (range 16–70 years).

Until the date of last contact seven (11%) patients are known to have died either of advanced DGC (*n* = 6) or LBC (*n* = 1) and one patient due to complications of liver cirrhosis. Mean age at death was 45 ± 14 years (range 21–72 years).

#### 2.4.1. Gastric Cancer

In 38 (61%) of the patients with a PV in *CDH1/CTNNA1* a DGC was detected either prior or after inclusion in this study with an age at diagnosis of 40 ± 13 years (range 16–72 years). Frequency of GC (26/40; 65% vs. 12/22; 55%; *p* = 0.419) and age at diagnosis (39.4 years vs. 42.8 years, *p* = 0.545) was similar between female and male patients. Overall 40 patients underwent gastrectomy with GC found in 78% (31/40). Eight patients received surgery after diagnosis of manifest GC. A total of 32 patients without a prior personal history of GC decided to undergo PTG. Subsequently, in 23 (72%) patients, SRCCs were detected. SRCCs found in PTG specimen were all confined to the mucosa (pT1a: *n* = 21; pTis: *n* = 2) and often multifocal (*n* = 11) with 1.6 (range 1–9) cancer foci per patient. The cancer foci were usually very small with a size of 0.2–2.5 mm in patients with PTG. A total of 15 patients opted to delay PTG out of personal or medical reasons or were too young. Extensive or metastatic disease was present in 15 (24%) patients with a mean age at diagnosis of 41 ± 14 years (range 16–72 years). These advanced DGCs presented predominantly as linitis plastica. No cancer foci were diagnosed in the duodenum (heterotopic gastric mucosa) or the esophagus (esophageal cardiac-type glands).

#### 2.4.2. Breast Cancer

Eight (20%) female patients belonging to seven families were diagnosed with LBC at a mean age of 51 ± 8 years (range 38–65 years). Six of them had family members with DGC and LBC. Four of them (with personal history of LBC) decided to undergo PTG, which revealed early (pT1a) SRCC in all. Two patients were classified as HLBC because family history lacked cases of DGC. One of them underwent PTG at age 39 and no SRCC was found.

### 2.5. Patients without a Proven CDH1 or CTNNA1 PV

In 140 patients out of 129 families we detected no PV in the *CDH1* gene (male 41; 29%, female 99; 71%). Thirty-eight patients presented with a family history of BC and GC but had no personal history of cancer. During the observed interval, five tumor-related deaths at a mean age of 42 ± 12 years (range 34–60 years) were reported in this group.

#### 2.5.1. Gastric Cancer

In 102/140 (73%) patients without PV GC was diagnosed (98 DGC; four intestinal-type gastric cancer) at a mean age of 44 ± 12 years (range 17–74 years). Cancer prevalence was 80% (33/41) in male and 70% (69/99) in female patients (*p* = 0.156). Mean age at cancer diagnosis was 45 ± 13 years (range 17–70 years) in male and 43 ± 12 years (range 18–74 years) in female patients (*p* = 0.352). Twenty-seven patients underwent gastrectomy after diagnosis of manifest cancer. Subsequently, all gastrectomy specimens showed GC. These patients typically exhibited a more advanced cancer stage upon histopathological staging ≥pT2 (*n* = 22), lymph node involvement (*n* = 11), and metastatic disease (*n* = 5). In only 5/27 (19%) patients was cancer confined to the gastric mucosa (pTis: *n* = 0; pT1a: *n* = 5). The remaining 75 patients all had advanced GC and were not eligible for surgery. As in the PV group, patients with advanced DGC typically showed a linitis plastica.

#### 2.5.2. Breast Cancer

Ten (10%) female patients were diagnosed with LBC at a mean age of 52 ± 7 years (range 45–65 years). Four had synchronous or metachronous DGC.

### 2.6. Comparison of PV Carriers with Patients without a PV

Comparing PV carriers with patients without a proven PV in the *CDH1* or *CTNNA1* genes showed no significant difference in gender ratio between those two groups (*p* = 0.38). Comparing patients without a PV and PV carriers, GC prevalence (73% vs. 61%; *p* = 0.1) and age at cancer diagnosis (44 ± 12 years vs. 40 ± 13 years; *p* = 0.255) were similar (Table 2, Figure 2). Again, there was no statistically significant difference in GC prevalence between patients without PV (102/129; 79%) and patients with PV (19/29; 66%) (*p* = 0.119) when comparing only index patients. Mean age at GC diagnosis in index patients without PV (44 ± 12 years; range 17–74 years) and index patients with PV (41 ± 12 years; range 16–61 years) was similar (*p* = 0.433)

Regarding LBC prevalence there was no significant difference between PV female carriers (20%) and female patients without a PV (10%) (*p* = 0.116). Average age of diagnosis was 51 ± 8 years and 52 ± 7 years, respectively (*p* = 0.923) (Figure 2). However, LBC prevalence in female index patients with PV (7/22; 32%) was significantly higher than in female index patients without PV (10/92; 11%) (*p* = 0.013).

### 2.7. Patients with a CDH1 VUS

Overall, five patients from three different families were diagnosed with a VUS in the *CDH1* gene (Table 4). GC prevalence was 40% (2/5) with a mean age at diagnosis of 41 ± 6 years (range 37–45 years). Family and personal history of patients with a *CDH1* VUS showed no cases of LBC.

## 3. Discussion

In this single center study, we report the results of genetic testing, surgery, and outcome in a large cohort of patients with suspected hereditary DGC and LBC, comparing carriers of a PV in *CDH1* or *CTNNA1* with patients in whom no underlying genetic cause could be identified.

Prevalence of *CDH1/CTNNA1* PVs in our cohort was 18% which is similar to studies from France which reported *CDH1* PVs in 19% and 14%, respectively, of patients meeting the 2015 HDGC testing criteria, whereas in a population with known founder mutations in New Zealand the detection rate of *CDH1* PVs was higher (40%) [31,32,37].

In our cohort, PV detection rate was highest in families meeting either criterion 3 or 1 (28% and 21%), resembling findings by Benusiglio and coworkers [31]. Age of onset of DGC in *CDH1* PV carriers varies between studies but is considered to be between 38 and 48 years [4,38,39]. Even though mean age of diagnosis in PV carriers was 40 years in our cohort, criterion 2 (one case of DGC before age 40) was least effective in detecting *CDH1/CTNNA1* PVs, which is in line with earlier studies [38,40]. In addition, we did not observe any statistically significant differences regarding age of onset of DGC between PV carriers and patients without PV. Hence, selection based on age of onset may not be appropriate to identify patients with suspected *CDH1/CTNNA1* PVs in similar populations.

HP infection is a significant risk factor for GC [41]. Although primarily associated with the development of intestinal-type GC, there are also reports suggesting HP infections to be linked to early-onset DGC, which might involve silencing of *CDH1*, *RUNX3*, *p21WAF,* and *p27* expression [42,43,44]. Additionally, familial clustering of HP infections has been described, indicating that HP might promote family aggregation of DGC [45,46]. However, a large Korean study showed no significant differences in HP seropositivity between patients with and without family history of GC [47]. In our cohort, we identified 28 (14%) patients with HP infection of whom 86% were diagnosed with GC. Among these were four patients without PV and with absence of any family history of DGC or LBC but early sporadic DGC. Therefore, it is tempting to speculate that HP infection might have been involved in establishing early onset DGC. Further studies are warranted to clarify this issue and to analyze the role of HP infection in DGC development in carriers of a *CDH1/CTNNA1* PV.

In our study, GC prevalence and age of onset did not differ significantly between PV carriers and patients without a PV. This might be due to a selection bias, as most of the patients without a PV underwent genetic counseling following a cancer diagnosis. In addition, family members of a *CDH1* or *CTNNA1* PV index patient, sought genetic and medical counseling, independent of personal cancer history, and were included in this study in case they were diagnosed with the same PV. However, even when only index patients were analyzed we did not observe any significant differences regarding GC prevalence and age of onset between the two groups.

Moreover, some cases of GC might have been missed in PV carriers, as 15 predictively tested patients with a proven PV opted to delay PTG and in 4/32 patients, who underwent PTG, histopathology was performed in hospitals without expertise in HDGC.

Cancer penetrance in PV carriers appears to vary between studies, among other reasons because of selection differences based on differing testing criteria [7,11,12]. A recent study showed that gastric cancer estimates in patients with a PV may be too high due to a selection bias [11].

The mean age at gastric cancer diagnosis was slightly higher in patients without germline PV than in PV carriers. However, those patients mostly presented with advanced cancer. In patients without a *CDH1/CTNNA1* PV only 27 underwent gastrectomy. In those, 81% (*n* = 22) had advanced GC. A majority (*n* = 75/102; 75%) of patients without PV were not considered for surgery due to advanced disease. In PV carriers, overall 40 patients underwent gastrectomy with 32 PTGs. In 31 gastrectomy specimens DGC was found. Unlike GCs in patients without PV, those were multifocal in 11 (35%) patients. 74% (*n* = 23) of SRCCs found in gastrectomy specimens were confined to the gastric mucosa. Comparing histopathological data from gastrectomy specimen, patients without a PV had significantly more advanced (≥pT2) GC than PV carriers (*p* = 0.00002), which might reflect intensified surveillance in PV families, thereby underlining the importance of developing improved surveillance endoscopy protocols in non-PV individuals and their first-degree relatives [7,48]. Alternatively, the molecular mechanisms underlying progression rate might differ between these groups [49,50].

We reassessed all CDH1 PVs according to the current guidelines published by the American College of Medical Genetics and Genomics and the Association for Molecular Pathology (ACMG/AMP-criteria) [33]. This led to the reclassification of the variant c.1108G > A; *p*.(Asp370Asn) to a VUS. In our final analysis we included this family in our PV group. Apart from the clear family history, the findings of the five family members included in our study were also very striking. All five underwent prophylactic gastrectomy. Three of them (ages 23, 41, 59) revealed SRCCs, in two cases multifocal.

A major challenge in clinical management of HDGC patients is presented by patients with a VUS. We identified five patients from three families with a strong family history or personal history of gastric cancer. Patients were advised to undergo regular endoscopic surveillance. Overall, DGC prevalence was 40%. Family history and personal history showed no cases of LBC. The age of onset of DGC was low in this subgroup.

LBC prevalence in patients with a proven PV was lower than previously reported [12]. Low LBC frequency in our cohort may be explained by our clinical focus on gastroenterology. Only 35% of patients with a PV fulfilled criterion 3 of the 2015 testing criteria (personal or family history of DGC and LBC with one diagnosed before age 50). We would therefore estimate that with the implementation of the novel 2020 criteria this number will rise as more patients with LBC will be included [7].

In our cohort we found six pathogenic germline variants that were present in more than one family (Table 2). All showed heterogeneity in the age of onset of manifest cancer. In three families the splice site variant c.1565+1G > A was detected. Two of those families reported high incidence of LBC but no cases of DGC. Remarkably, the pedigree of the third family did not include cases of LBC but four family members with DGC. Thus, the heterogeneity of the manifestation for the same PV of the CDH1 gene suggests that exogenous factors (smoking, alcohol abuse), endogenous factors (microbiom, hormones), or polygenic causes could be involved in the etiology of the different clinical presentations. Further research in large, well characterized cohorts is necessary.

As already described in other hereditary tumor syndromes, type and location of a germline PV on the *CDH1* gene may influence penetrance [40,51,52,53]. However, the aforementioned cases illustrate the heterogeneity in expression of the same variants as well as a potential influence of yet unidentified factors on carcinogenesis in carriers of a pathogenic *CDH1* or *CTNNA1* variant. In currently valid and used guidelines the clinical management is heavily dependent on family history, especially in the case of HLBC where PTG is no longer recommended but can be considered [7]. However, patients must be interviewed in detail as extended family history might reveal cases of DGC in these families. These reports from our cohort show that genotype–phenotype correlation of those variants is unclear, and management based on family history could potentially lead to incorrect treatment.

There are several limitations to the design of this study. In a majority of patients only single-site *CDH1* testing was performed. Hence, large scale sequencing studies, in particular whole exome approaches in this cohort may identify additional novel causative genes. Since we did not implement the *CTNNA1* gene into our diagnostics until August 2018, patients may have been incorrectly assigned to the non-PV group. However, it should be noted that the detection rate of PV of *CTNNA1* gene is not very high in patients who meet the 2015 IGCLC criteria [29].

Another issue is patients lost during follow-up, especially patients without a *CDH1* or *CTNNA1* PV. This will require further investigation, especially regarding the recommended surveillance examinations in these patients. Given the number of cases of advanced cancer in this cohort we estimate that most patients have died in the meantime.

## 4. Materials and Methods

### 4.1. Study Population

Individuals with DGC and or positive family history were referred for genetic consultation at the national center for hereditary tumor syndromes (NZET) from January 2016 to August 2020. Patients were eligible for this register study if they fulfilled the established 2015 IGCLC testing criteria or the extended criteria, where testing can be considered [6]. Relatives of patients with a proven PV that underwent genetic testing and were tested negative for a *CDH1/CTNNA1* PV were excluded. We defined the index patient as the first family member who fulfilled the HDGC testing criteria and underwent genetic testing. This registry study was approved by the ethics committee (Ethics committee of the University Hospital Bonn, approved 14 May 2015, No. 099/15). Signed informed consent was obtained from all patients.

Patients with a pathogenic or likely pathogenic variant who opted for further treatment at NZET were referred to a multidisciplinary team of gastroenterologists, gastrointestinal oncologists, surgeons, and geneticists. Counselling was performed following current HDGC guidelines. Preoperative endoscopy or surveillance endoscopy was conducted by a team of trained physicians and gastrectomy specimens were examined by a designated gastrointestinal pathologist. For participants who were not treated at our hospital all relevant available medical records including 3-generation (or more) family pedigrees, genetic test results, and endoscopic, surgical, and pathology reports were reviewed. Parameters at baseline included date of birth, gender, personal and family cancer history as well as data from genetic analysis, preoperative endoscopy, and surgical reports. If possible, we collected follow-up data from surveillance endoscopies, surgeries, and imaging reports. For deceased patients we attempted to evaluate the probable cause of death.

Patients without a (likely-)pathogenic variant or carriers of a VUS were advised to undergo annual endoscopic surveillance.

### 4.2. Genetic Testing

Patients with a family history of DGC and/or LBC but without prior diagnosis of a PV in *CDH1* or *CTNNA1* received genetic counseling. In patients fulfilling the 2015 HDGC testing criteria, Sanger sequencing of the *CDH1* gene including the intron–exon boundaries or next generation sequencing based multigene panel (customized version of the TruSight™Cancer Sequencing Panel including *CTNNA1*; Illumina, San Diego, CA, USA) was performed using leucocyte DNA. Multiplex ligation-dependent probe amplification (MLPA) was used to detect large genomic rearrangements (MRC Holland, Amsterdam, The Netherlands). All patients presenting after August 2018 received sequencing of the *CTNNA1* gene, in case no *CDH1* PV had been detected. Interpretation of variants was based on the American College of Medical Genetics and Genomics and the Association for Molecular Pathology guidelines [54]. All variants were later assessed according to the current ACMG-AMP classification of *CDH1* variants [33].

### 4.3. Data Analysis

Data were compiled into a RedCap (www.project-redcap.org) database developed for this project. Data retrieval and statistical analysis was performed using SPSS (Version 27, https://www.ibm.com/analytics/spss-statistics-software). Continuous variables are shown as mean ± standard deviation and range. Variance equality between two groups was assessed using Levene’s test. For continuous variables Student’s *t*-test and Welch’s *t*-test were used to determine differences between the groups. Pearson’s chi-square test was used to compare categorical variables. A *p* value < 0.05 was considered significant.

## 5. Conclusions

When using the 2015 HDGC testing criteria the detection rate of *CDH1/CTNNA1* PVs was similar to previously described cohorts. The prevalence of DGC and LBC varied between families with a proven germline variant. Occurrence of DGC in PV carriers was similar to previous studies. Although more female variant carriers were included, we reported fewer cases of LBC in patients with a *CDH1/CTNNA1* PV than other studies. The comparison of patients without a PV with PV carriers exhibits a similar LBC and DGC prevalence, and a similar age of onset. Most PV carriers were diagnosed with early DGC in PTG, whereas most patients without PV had advanced cancer. Since until now most patients have no identifiable cause for familial aggregation of early DGC and LBC, large scale sequencing studies are needed to identify novel causative genes. Further investigation will show if clinical guidelines can improve the outcome for these patients.

## Figures and Tables

**Figure 1 cancers-12-03726-f001:**
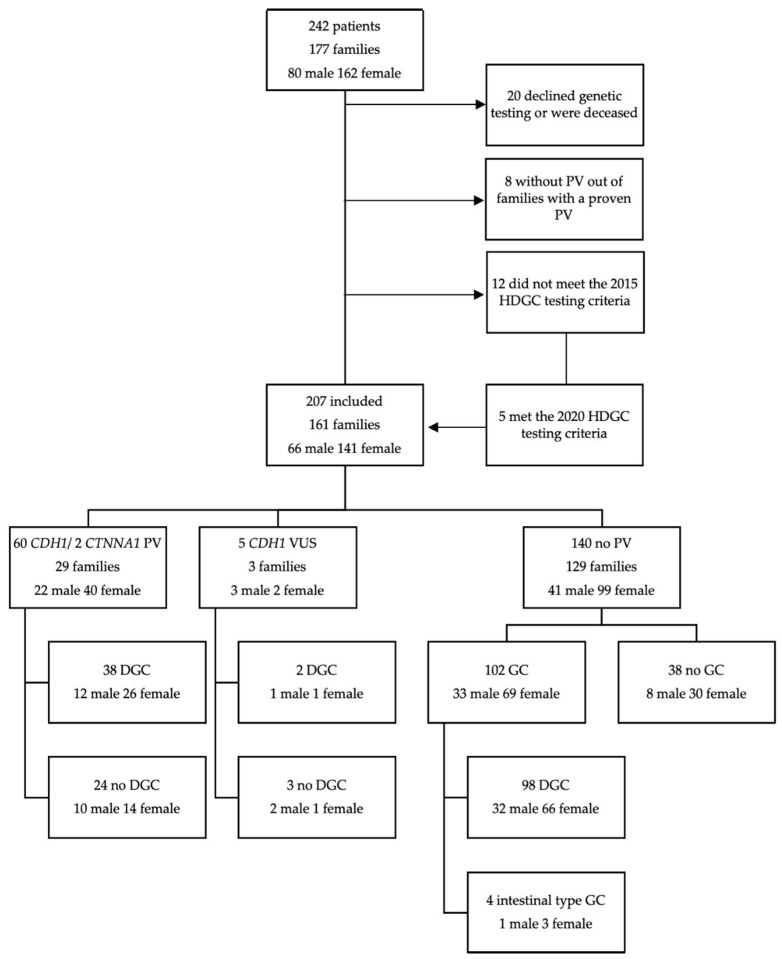
Study population and gastric cancer incidence. (PV= (likely-) pathogenic variant, HDGC = hereditary diffuse gastric cancer, VUS = variant of unknown significance, DGC = diffuse gastric cancer, GC = gastric cancer).

**Figure 2 cancers-12-03726-f002:**
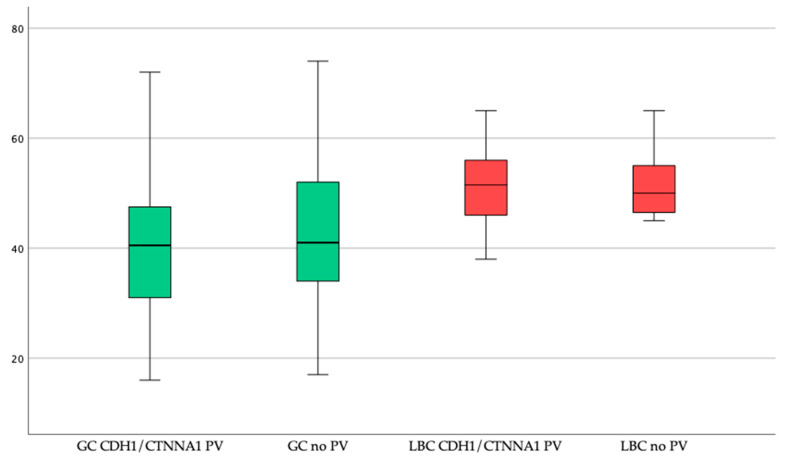
Age at diagnosis of gastric cancer (GC) and lobular breast cancer (LBC) in patients with a *CDH1/CTNNA1* pathogenic or likely pathogenic variant (PV) and without PV.

**Table 1 cancers-12-03726-t001:** HDGC testing criteria from 2015 and 2020.

HDGC Testing Criteria from 2015 [6]	HDGC Testing Criteria from 2020 [7]
Established testing criteria	Family criteria
Two cases of gastric cancer regardless of age with at least one confirmed DGC One case of DGC before age 40 Personal or family history of DGC and LBC with one diagnosed before age 50	Two cases of gastric cancer regardless of age with at least one confirmed DGC One case of DGC at any age, and one case of LBC before age 70 in different family members Two cases of LBC before age 50
Extended criteria (testing could be considered)	Individual criteria
Bilateral LBC or family history of two or more cases of LBC before age 50 Personal or family history of cleft lip/palate in patients with DGC In situ signet ring cells and/or pagetoid spread of signet ring cells	DGC before age 50 DGC at any age in individuals of Maori ethnicity DGC in individuals with a personal or family history of cleft lip/palate DGC and LBC before age 70 Bilateral LBC before age 70 In situ SRCC before age 50

(HDGC = hereditary diffuse gastric cancer, DGC = diffuse gastric cancer, LBC = lobular breast cancer, SRCC = signet ring cell carcinoma).

**Table 2 cancers-12-03726-t002:** Patient demographics, cancer prevalence, findings in gastrectomy (PV = (likely-) pathogenic variant; VUS = variant of unknown significance; PTG = prophylactic total gastrectomy *, GC = gastric cancer, LBC = lobular breast cancer).

Baseline and Clinical Characteristics	*CDH1/CTNNA1* PV	*CDH1* VUS	No PV
Number of patients	62	5	140
Gender	22/62 (35%) male 40/62 (65%) female	3/5 (60%) male 2/5 (40%) female	41/140 (29%) male 99/140 (71%) female
Number of families	29	3	129
Age at study inclusion (range)	42 ± 14 years (16–70)	49 ± 8 years (39–62)	49 ± 15 years (18–87)
GC prevalence	38/62 (61%)	2/5 (40%)	102/140 (73%)
Age at GC diagnosis (range)	40 ± 13 years (16–72)	41 ± 6 years (37–45)	44 ± 12 years (17–74)
LBC prevalence in women	8/40 (20%)	0	10/99 (10%)
Age at LBC diagnosis (range)	51 ± 8 years (38–65)	–	52 ± 7 years (45–65)
Number of gastrectomies (PTG *)	40 32 *	1 0 *	27 0 *
Yield of GC in gastrectomies (PTG *)	31/40 (78%) 23/32 (72%) *	1/1 (100%)	27/27 (100%)
Cancer staging in gastrectomy <pT2 ≥pT2	23/31 (74%) 8/31 (26%)	0/1 (0%) 1/1 (100%)	5/27 (19%) 22/27 (81%)
Patients with multifocal GC in gastrectomy	11/31 (35%)	0/1 (0%)	0/27 (0%)

**Table 3 cancers-12-03726-t003:** Index patients fulfilling the HDGC testing criteria (2015) and detection rate of pathogenic *CDH1/CTNNA1* variants (PV = (likely-) pathogenic variant, DGC = diffuse gastric cancer, LBC = lobular breast cancer).

2015 HDGC Criteria	Index Patients	Index Patients with *CDH1/CTNNA1* PV	Detection Rate
Criterion 1 (Two cases of gastric cancer regardless of age with at least one confirmed DGC)	97	20	21%
Criterion 2 (One case of DGC before age 40)	87	13	15%
Criterion 3 (Personal or family history of DGC and LBC with one diagnosed before age 50)	40	11	28%
Patients fulfilling more than one criterion	66	15	23%
Total (index patients)	161	29	18%
Total (all patients)	207	62	30%

**Table 4 cancers-12-03726-t004:** Detected (likely) pathogenic germline variants (PV) in *CDH1* and *CTNNA1* genes, as well as variants of unknown significance (VUS) in the CDH1 gene, according to pedigree data (except the last PV, all variants are located in the *CDH1* gene) [33,34,35,36].

Variant	Reported Variant Classification	Classification with CDH1-Specific Criteria [33]	Type of Variant	Number of Family Members Included	Reported GC in Family Pedigree	Reported BC in Family Pedigree	Mean Age at Diagnosis of Manifest Cancer *
(Likely) pathogenic variants							
c.3G>A;p.?	pathogenic	pathogenic	start lost	6	2	0	47 (46–47)
c.48+1G>A;p.?	pathogenic	pathogenic	splice	3	4	0	52 (33–75)
c.86dupA;p.(His296Glnfs*5)	pathogenic	likely pathogenic	frameshift	1	2	1	29 (26–35)
c.521delA;p.(Asn174Thrfs*41) mosaicism	pathogenic	likely pathogenic	frameshift	1	1	1	41 (41)
c.1108G>A;p.(Asp370Asn)	likely pathogenic	VUS	missense	5	4	0	48 (41–61)
c.1137G>A;p.?	pathogenic	pathogenic	splice [34]	1	4	0	47 (44–50)
c.1416dupC;p.(Val473Argfs*10)	pathogenic	likely pathogenic	frameshift	1	1	0	48 (48)
c.1565+1G>A;p.?	pathogenic	pathogenic	splice	1	0	2	49 (38–60)
c.1565+1G>A;p.?	pathogenic	pathogenic	splice	2	4	0	47 (32–64)
c.1565+1G>A;p.?	pathogenic	pathogenic	splice	1	0	5	65 (65)
c.1565+2dupT;p.?	likely pathogenic	pathogenic	splice	3	1	0	34 (34)
c.1565+2dupT;p.?	likely pathogenic	pathogenic	splice	1	2	0	50 (50)
c.1651G>T;p.(Glu551*)	pathogenic	likely pathogenic	nonsense	3	3	1	55 (40–76)
c.1679C>G;r.1680_1711del;p.(Tyr561Phefs*16)	pathogenic	pathogenic	splice [35]	1	2	1	49 (47–50)
c.1679C>G;r.1680_1711del;p.(Tyr561Phefs*16)	pathogenic	pathogenic	splice [35]	1	3	0	unknown
c.1746_1747dup;p.(Leu583Argfs*2)	pathogenic	likely pathogenic	frameshift	3	4	0	unknown
c.1746_1747dup;p.(Leu583Argfs*2)	pathogenic	likely pathogenic	frameshift	4	6	1	unknown
c.1786G>T;p.(Glu596*)	pathogenic	likely pathogenic	nonsense	1	2	0	46 (44–47)
c.1792C>T;p.(Arg598*)	pathogenic	pathogenic	nonsense	1	3	0	30 (23–37)
c.1792C>T;p.(Arg598*)	pathogenic	pathogenic	nonsense	2	4	1	51 (43–63)
c.1901C>T;r.1900_1936del; p.(Ala634Profs*7)	pathogenic	pathogenic	splice [36]	2	3	2	16 (16)
c.2116C>T;p.(Gln706*)	pathogenic	likely pathogenic	nonsense	4	3	2	53 (52–54)
c.2165-2A>C;p?	pathogenic	likely pathogenic	splice	3	2	0	40 (32–48)
deletion exon 1–2	pathogenic	pathogenic	large exonic deletion	1	2	0	39 (33–45)
deletion exon 1–2	pathogenic	pathogenic	large exonic deletion	1	4	0	43 (32–49)
deletion exon 3–8	pathogenic	likely pathogenic	large exonic deletion	1	3	2	49 (38–58)
deletion exon 1–16	pathogenic	pathogenic	large exonic deletion	1	2	0	47 (37–56)
deletion exon 8–11	pathogenic	likely pathogenic	large exonic deletion	5	7	5	56 (48–72)
*CTNNA1*: c.1175delA;p.(Asp392Valfs*13)	pathogenic	likely pathogenic	frameshift	2	2	0	44 (44)
Variants of uncertain significance (VUS)							
c.659T>G;p.(Leu220Arg)	VUS	VUS	missense	2	2	0	47 (45–49)
c.1466C>T;p.(Pro489Leu)	VUS	VUS	missense	1	1	0	37 (37)
c.2629G>A; p.(Gly877Arg)	VUS	VUS	missense	2	2	0	43 (32–54)

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
