# Peer review of "Hereditary Diffuse Gastric Cancer: A Comparative Cohort Study According to Pathogenic Variant Status"

_cancers, 2020, doi:10.3390/cancers12123726_

Round 1
Reviewer 1 Report
Overall, and despite that it still lacks critical discussion of results, the manuscript has significantly improved.
Minor revisions:
- Table 4 caption should be corrected. I suggest:
Table 4. Detected (likely) pathogenic germline variants (PV) in CDH1 and CTNNA1 genes, as well as CDH1 VUS, according to pedigree data (except for the last variant, all variants are located in the CDH1 gene)
- Lines 213 and 295: The authors write “CDH1 or CTNNA1 gene”, which should be altered to “CDH1 or CTNNA1 genes”.
Author Response
Overall, and despite that it still lacks critical discussion of results, the manuscript has significantly improved.
Minor revisions:
Table 4 caption should be corrected. I suggest:
Table 4. Detected (likely) pathogenic germline variants (PV) in CDH1 and CTNNA1 genes, as well as CDH1 VUS, according to pedigree data (except for the last variant, all variants are located in the CDH1 gene)
Response:
We thank the reviewer for this comment. We changed the manuscript accordingly (line 238-240)
Lines 213 and 295: The authors write “CDH1 or CTNNA1 gene”, which should be altered to “CDH1 or CTNNA1 genes”.
Response:
We apologize for this mistake and have updated the manuscript.
Reviewer 2 Report
The authors have carefully addressed all points raised during revision, demonstrating a great effort to improve the manuscript and a strong commitment with results presented. The article is now suitable for publication.
I would like to reinforce that based on author’s findings, 21 different CDH1 pathogenic variants + 1 CTNNA1 PV could be considered (instead of 20 + 1, referred in the abstract and in section 2.4).
Author Response
The authors have carefully addressed all points raised during revision, demonstrating a great effort to improve the manuscript and a strong commitment with results presented. The article is now suitable for publication.
I would like to reinforce that based on author’s findings, 21 different CDH1 pathogenic variants+ 1 CTNNA1 PV could be considered (instead of 20 + 1, referred in the abstract and in section 2.4).
Response:
We thank the reviewer for this valuable comment and apologize for this mistake. We changed the manuscript accordingly (21 CDH1 PV + 1 CTNNA1 PV).
Reviewer 3 Report
I think the work has improved substantially with the changes made.
As a minor comment, I think that the authors should include in the material and methods section their criteria for defining the index case. In my experience, most groups look for the index case among the subjects (within the family who meet criteria for hereditary cancer) who have developed the disease.
Author Response
I think the work has improved substantially with the changes made.
As a minor comment, I think that the authors should include in the material and methods section their criteria for defining the index case. In my experience, most groups look for the index case among the subjects (within the family who meet criteria for hereditary cancer) who have developed the disease.
Response:
We thank the reviewer for this valuable comment. We added our definition of the index patient to the manuscript (line 427-429)
This manuscript is a resubmission of an earlier submission. The following is a list of the peer review reports and author responses from that submission.
Round 1
Reviewer 1 Report
This well written paper is a thorough and most valuable description of a large HDGC cohort. A number of new CDH1 mutations have been described, providing a valuable reference source to help validate new mutations in other families.
I have a small number of minor concerns:
line 51-53: The 1998 paper that is referred to identified three different CDH1 in three separate Maori families (not one).
line 314: confound-> confined
Author Response
This well written paper is a thorough and most valuable description of a large HDGC cohort. A number of new CDH1 mutations have been described, providing a valuable reference source to help validate new mutations in other families.
I have a small number of minor concerns:
line 51-53: The 1998 paper that is referred to identified three different CDH1 in three separate Maori families (not one).
Response:
Thank you very much for your valuable input. We have corrected line 51-53.
line 314: confound-> confined
Response:
We apologize for this mistake that has been corrected in the revised version of our manuscript.
Reviewer 2 Report
In this study, authors report results of genetic testing, surgery, and outcome in a large cohort of families meeting the 2015 testing criteria for hereditary diffuse gastric cancer. The aim is to compare carriers with a pathogenic variant mutation in CDH1 or CTNNA1 with patients in whom no underlying cause has been identified yet.
The paper is original and interesting, and the overall presentation is appropriate, but I request some clarification
In this study, authors report results of genetic testing, surgery, and outcome in a large cohort of families meeting the 2015 testing criteria for hereditary diffuse gastric cancer. The aim is to compare carriers with a pathogenic variant mutation in CDH1 or CTNNA1 with patients in whom no underlying cause has been identified yet.
The paper is original and interesting, and the overall presentation is appropriate, but I request some clarification:
1. The study population, as described in Figure 1, is confusing, and requires clarification by the authors.
I assume that subjects from PV families who are not carriers of the mutation are not included in the study, as their risk is assumed to be low; that is, two groups of patients whose families meet criteria for the genetic diagnosis of HDGC are compared: a group corresponding to subjects with a known germline mutation and another group of families in which all members are at risk, since no PV has been detected in their families.
2. In this case, I have two comments on which I would appreciate clarification from the authors. 1) the authors do not explain the follow-up that non-PV carriers have received (Methods section), and 2) If the prevalence of GC is similar in both groups, this indicates that the risk is much higher in the families in which no PV is detected, since in genetically defined families only 50% of individuals inherit the risk associated with the mutation in an autosomal dominant manner.
3. Results, line 118-120: sequencing of the CDH1 gen was done in the whole cohort, but CTNNA1 gene was additionally analyzed only in a subset of patients. Authors should indicate in how many patients CTNNA1 gene was analyzed. If this gene was not studied, some PV patients may have been assigned to the wrong group.
4. Results, line 122-124: “Nine patients were diagnosed with a variant of uncertain significance.” Their status may be different from those diagnosed with benign variants or without variants. Please explain if these patients were excluded from the study or became part of the “no PV” group.
5. Results, line 227-30; 15 patients opted to delay PTG, and extensive or metastatic disease was present in 15. Are these figures ok? How long did it take from genetic diagnosis to advanced cancer diagnosis?
6. Discussion, line 275-6: Based on their results, the authors cannot state: “… that the majority of CDH1 PV carriers are not being identified when patients are selected according to the established clinical criteria only” (this statement is based solely in bibliographic resources)
7. Discussion, line294-5, and results paragraph 4.2: “to reduce the influence of a selection bias, we compared the GC prevalence in the index patients of the families” Who were the index patients in each group (with or without PV)?
Author Response
In this study, authors report results of genetic testing, surgery, and outcome in a large cohort of families meeting the 2015 testing criteria for hereditary diffuse gastric cancer. The aim is to compare carriers with a pathogenic variant mutation in CDH1 or CTNNA1 with patients in whom no underlying cause has been identified yet.
The paper is original and interesting, and the overall presentation is appropriate, but I request some clarification
In this study, authors report results of genetic testing, surgery, and outcome in a large cohort of families meeting the 2015 testing criteria for hereditary diffuse gastric cancer. The aim is to compare carriers with a pathogenic variant mutation in CDH1 or CTNNA1 with patients in whom no underlying cause has been identified yet.
The paper is original and interesting, and the overall presentation is appropriate, but I request some clarification:
1. The study population, as described in Figure 1, is confusing, and requires clarification by the authors.
I assume that subjects from PV families who are not carriers of the mutation are not included in the study, as their risk is assumed to be low; that is, two groups of patients whose families meet criteria for the genetic diagnosis of HDGC are compared: a group corresponding to subjects with a known germline mutation and another group of families in which all members are at risk, since no PV has been detected in their families.
Response:
We thank the reviewer for this important point. In the revised version of the manuscript we have excluded eight non-PV carriers from PV families, as well as all patients who refused genetic testing (results, lines 115-122). Furthermore, we have updated figure 1 to be clearer.
In this case, I have two comments on which I would appreciate clarification from the authors. 1) the authors do not explain the follow-up that non-PV carriers have received (Methods section)
Response:
We agree that this was not properly adressed. Patients were advised to undergo annual endoscopic examination. This is mentioned in the revised methods section (line 470-471).
, and 2) If the prevalence of GC is similar in both groups, this indicates that the risk is much higher in the families in which no PV is detected, since in genetically defined families only 50% of individuals inherit the risk associated with the mutation in an autosomal dominant manner.
Response:
We kindly disagree with this observation. To elucidate the equal GC risk of both groups, we included the analysis of GC prevalence in index patients. We observed no signicant difference in GC prevalence between patients without PV (79%) and patients with PV (66%). This is explained in the results section (line 296-303). Furthermore, GC risk in proven PV patients might also be underestimated, because not all PV patients underwent prophylactic gastrectomy or gastrectomy specimen were not thoroughly inspected (discussed in line 376-381).
Results, line 118-120: sequencing of the CDH1 gen was done in the whole cohort, but CTNNA1 gene was additionally analyzed only in a subset of patients. Authors should indicate in how many patients CTNNA1 gene was analyzed. If this gene was not studied, some PV patients may have been assigned to the wrong group.
Response:
We agree with the reviewer that this is of specific interest. We have clarified this issue in the results/discussion section (results line 121-123; discussion 440-444).
We agree that some non-PV patients might have been assigned to the wrong group. But it has to be mentioned that detection rate of pathogenic CTNNA1 variants in patients fulfilling the IGLC-criteria (version of 2015) is fairly low as published by Benusiglio et al., where a detection of 1/32 patients was documented (Benusiglio et al Gastric Cancer. 2019 Jul;22(4):899-903.doi: 10.1007/s10120-018-00907-7. Epub 2018 Dec 4.).
Results, line 122-124: “Nine patients were diagnosed with a variant of uncertain significance.” Their status may be different from those diagnosed with benign variants or without variants.
Please explain if these patients were excluded from the study or became part of the “no PV” group.
Response:
The reviewer raises a very important point as VUS possess a major challenge. We have reassessed all CDH1 variants according tot he ACMG/AMG criteria by Lee et al.. This led to a reclassification of VUS. In our final analysis we included 5 patients out of 3 families. This patient group was not excluded and was analysed seperately (table 2 and results section line 313-316).
Results, line 227-30; 15 patients opted to delay PTG, and extensive or metastatic disease was present in 15. Are these figures ok? How long did it take from genetic diagnosis to advanced cancer diagnosis?
Response:
We agree with the reviewer, that this section is misleadingly worded. We have clarified this issue in the revised version (line 257-260). Metastatic disease was apparent at the time of initiation of germline testing in 15 patients. A different set of 15 patients opted to delay PTG.
Discussion, line 275-6: Based on their results, the authors cannot state: “… that the majority of CDH1 PV carriers are not being identified when patients are selected according to the established clinical criteria only” (this statement is based solely in bibliographic resources)
Response:
We agree with this comment and have revised the manuscript accordingly (discussion lines 340-341).
Discussion, line294-5, and results paragraph 4.2: “to reduce the influence of a selection bias, we compared the GC prevalence in the index patients of the families” Who were the index patients in each group (with or without PV)?
Response:
The index case of each family is the case of the original patient who fulfilled the criteria for further testing of CDH1/CTNNA1.
Reviewer 3 Report
In this manuscript, the authors aim to determine the contribution of CDH1/CTNNA1 mutations to the outcome of HDGC patients. For that purpose, the group has performed a comparative study between HDGC patients carriers and non-carriers of pathogenic CDH1/CTNNA1 mutations. A large cohort of 230 patients, encompassing 163 HDGC families, was assembled and evaluated regarding gastric cancer prevalence, age at diagnosis, gastrectomy findings and occurrence of lobular breast cancer, among other parameters. This is indeed the major strength of the work. However, the way the results are presented and their poor description impair a clear analysis and interpretation of the data. Overall, the article suffers from some critical points that should be addressed for the manuscript to be published.
Main comments:
- In this study, selected families met the HDGC testing criteria from 2015. However, the latest HDGC guidelines were recently published (August 2020) and include different genetic testing criteria. Why were the analysis performed based on outdated criteria? As referred by the authors, additional patients could be enrolled in this work according to the 2020 updated guidelines. Still, those patients were not included for further analysis. Why were these patients not included?
- Despite the authors claim to have investigated the data from 230 patients, only 210 were screened for CDH1/CTNNA1 genetic alterations. 20 patients are misinterpreted. This should be corrected along the analysis and manuscript.
- In Table 2, the vast majority of displayed percentages are not clear. It is hard to figure out how these numbers were calculated since the total number is not the same. This should be improved. The same happens in table 3.
- Statistical analysis should be provided concerning the data presented in the tables.
- The authors stated that the criterion 3 (personal or family history of DGC and LBC with one diagnosed before age 50) yielded the highest detection rate of CDH1 and CTNNA1 pathogenic variants. This should be clarified since criterion 3 was able to select 22 patients out of the 62 carriers, when compared with 49 selected patients from the criterion 1 or 28 from criterion 2. Authors should be cautious with the way the results are presented.
- The authors stated that H. pylori infection was detected in 30 patients. Nevertheless, no information is provided regarding the methods used to identify this pathogen.
- Table 4 summarizes the data associated with the 29 germline pathogenic variants identified in CDH1/CTNNA1 genes. A table displaying the remaining germline variants (classified as benign, likely benign and VUS) and respective clinical data should be provided. Further, variant classification was performed according to the ACMG/AMP guidelines. Were the authors aware of the specifications of the ACMG/AMP variant curation guidelines for the analysis of germline CDH1 sequence variants?
- Gastric cancer histopathological features were very briefly described but the data is not presented.
- In the discussion section, the authors highlight the heterogeneity in age of onset and in clinical manifestations of families that carry the same genetic alteration. This is a very interesting point, although no possible explanations are discussed by the team. A decontextualized comment suggesting that the type and location of CDH1 pathogenic variants may influence penetrance is provided instead.
- Results should be described in a more comprehensive and detailed way, properly referring tables/figures. In the present form, tables/figures are referenced too late making hard to follow the results.
- “the underlying mechanism for most of the patients with a family or personal history of DGC and LBC has so far not been elucidated” – please specify the percentages lacking a causative alteration. The same should be clarified in “CDH1 PVs are only present in a minority of those patients who fulfil the testing criteria”.
Author Response
In this manuscript, the authors aim to determine the contribution of CDH1/CTNNA1 mutations to the outcome of HDGC patients. For that purpose, the group has performed a comparative study between HDGC patients carriers and non-carriers of pathogenic CDH1/CTNNA1 mutations. A large cohort of 230 patients, encompassing 163 HDGC families, was assembled and evaluated regarding gastric cancer prevalence, age at diagnosis, gastrectomy findings and occurrence of lobular breast cancer, among other parameters. This is indeed the major strength of the work. However, the way the results are presented and their poor description impair a clear analysis and interpretation of the data. Overall, the article suffers from some critical points that should be addressed for the manuscript to be published.
Main comments:
- In this study, selected families met the HDGC testing criteria from 2015. However, the latest HDGC guidelines were recently published (August 2020) and include different genetic testing criteria. Why were the analysis performed based on outdated criteria? As referred by the authors, additional patients could be enrolled in this work according to the 2020 updated guidelines. Still, those patients were not included for further analysis. Why were these patients not included?
Response:
We thank the reviewer for this justified criticism. We have included patients who do not meet the 2015 criteria but who meet the 2020 criteria in our final analysis. Unfortunately, we cannot retrospectively change the inclusion criteria because no genetic diagnostics were performed if the criteria of 2015 were not met. This is now also discussed in the manuscript (discussion lines 337-341).
- Despite the authors claim to have investigated the data from 230 patients, only 210 were screened for CDH1/CTNNA1 genetic alterations. 20 patients are misinterpreted. This should be corrected along the analysis and manuscript.
Response:
We completely agree with the reviewer and have revised our manuscript accordingly (results lines 115-121).
- In Table 2, the vast majority of displayed percentages are not clear. It is hard to figure out how these numbers were calculated since the total number is not the same. This should be improved. The same happens in table 3.
Response:
We have clarified this issue and have revised table 2 and 3.
- Statistical analysis should be provided concerning the data presented in the tables.
Response:
Stastitical analysis is included in the results section. We did not include them in the tables because we have included a separate VUS-section.
- The authors stated that the criterion 3 (personal or family history of DGC and LBC with one diagnosed before age 50) yielded the highest detection rate of CDH1 and CTNNA1 pathogenic variants. This should be clarified since criterion 3 was able to select 22 patients out of the 62 carriers, when compared with 49 selected patients from the criterion 1 or 28 from criterion 2. Authors should be cautious with the way the results are presented.
Response:
We thank the reviewer for this comment. We revised the results, table 3 and the discussion concerning the HDGC criteria. According to Benusiglio et al and Aronson et al, the detection rate of each criterion is defined as the percentage of index patients with a CDH1/CTNNA1 PV of all index patients meeting the specific criterion.
- The authors stated that H. pylori infection was detected in 30 patients. Nevertheless, no information is provided regarding the methods used to identify this pathogen.
Response:
We have adressed this issue in the revised results section. Diagnosis of H. pylori infection was made by histopathological analysis.
- Table 4 summarizes the data associated with the 29 germline pathogenic variants identified in CDH1/CTNNA1 genes. A table displaying the remaining germline variants (classified as benign, likely benign and VUS) and respective clinical data should be provided. Further, variant classification was performed according to the ACMG/AMP guidelines. Were the authors aware of the specifications of the ACMG/AMP variant curation guidelines for the analysis of germline CDH1 sequence variants?
Response:
Thank you for this valuable comment. We agree that especially VUS are of certain interest. We have included a separate VUS section in table 2 and 3. Furthermore, we have included VUS in table 4. However, we decided not to include the (likely) benign variants, because those were given only in former reports and not in the reports of the recent years, since it is not usual anymore and gives no additional relevant information. After your comment we realised, that we have used the ACMG/AMP guidelines only for the most recent variants. We were aware of the CDH1-specific guidelines by Lee et al. And decided to reclassify all variants based on these guidelines. The use of these guidelines resulted in some variants in the classification of „likely pathogenic“ instead of „pathogenic“, because only criteria PVS1 and PM2 are fulfilled. In the original diagnostic reports it was usual to use the presence of a truncating variant in the tumor suppressor gene CDH1 in a patient with fulfilled clinical criteria as pathogenic. In one patient a missense variant, which was originallyreported as likely pathogenic, resulted in the classification as VUS. However, we decided to leave this family in the cohort with identified CDH1 variants, because the family history is very suitable for HDGC (index patient diffuse gastric cancer at age 58, sister stomach cancer at age 45 (deceased), maternal aunt stomach cancer at age 42 (deceased)). Apart from the clear family history, also the findings of the five family members included in our study were very striking. All five underwent prophylactic gastrectomy. Three of them (age 23, 41, 59) revealed SRCCs, in two cases multifocal. This is discussed in the discussion section lines 401-405.
- Gastric cancer histopathological features were very briefly described but the data is not presented.
Response:
In the revised manuscript we thoroughly describe gastric cancer in the PV and non-PV group according to age of onset, cancer stage and number of cancer foci and typical histopathological features (lines 250-261 and lines 279-287).
- In the discussion section, the authors highlight the heterogeneity in age of onset and in clinical manifestations of families that carry the same genetic alteration. This is a very interesting point, although no possible explanations are discussed by the team. A decontextualized comment suggesting that the type and location of CDH1 pathogenic variants may influence penetrance is provided instead.
Response:
We agree with this important comment and have revised this section (lines 423-426). Possible causes could be exogenous factors (smoking, alcohol abuse), endogenous factors (microbiom, hormones) or polygenic causes. Further research in large, well characterized cohorts is necessary.
- Results should be described in a more comprehensive and detailed way, properly referring tables/figures. In the present form, tables/figures are referenced too late making hard to follow the results.
Response:
We have updated the manuscipt.
- “the underlying mechanism for most of the patients with a family or personal history of DGC and LBC has so far not been elucidated” – please specify the percentages lacking a causative alteration.
Response:
We thank the reviewer for this comment. We updated the manuscript and added the percentages from studies, cited in the introduction.
The same should be clarified in “CDH1 PVs are only present in a minority of those patients who fulfil the testing criteria”.
Response:
We added another reference (Benusiglio et al) and stated the percentages of patients with CDH1 PVs in those cohorts.
Reviewer 4 Report
This manuscript describes germline mutation of CDH1 and CTNNA1 genes in hereditary diffuse cancer in German population. This is interesting results.
1. Author described that genetic testing identified 60 carriers of a pathogenic or likely pathogenic CDH1 germline variant out of 28 families. The total number of carriers with CDH1/CTNNA1 pathogenic variant described in Table 3 have been exceed 60 carriers. Author should be excluded overlapping carriers.
2. Author performed sequencing of CDH1 and CTNNA1 genes for 210 of 230 patient. In table 2, the total number of patients of CDH1/CTNNA1 pathogenic variant and no pathogenic variant was 230 patients. Author should be excluded 20 patients that declined genetic testing or died before testing was conducted.
3. Although author diagnosed helicobacter pylori infection for 30 patients, helicobacter pylori infection should diagnose for all gastric cancer patients.
4. Author identified some splice site variants in this study. Author should confirm by RT-PCR and sequencing analysis whether these variants cause splicing abnormalities.
5. Page 9, line 287-290 and 325-328, these sentence and figure 2 should described in the results.
Author Response
This manuscript describes germline mutation of CDH1 and CTNNA1 genes in hereditary diffuse cancer in German population. This is interesting results.
- Author described that genetic testing identified 60 carriers of a pathogenic or likely pathogenic CDH1 germline variant out of 28 families. The total number of carriers with CDH1/CTNNA1 pathogenic variant described in Table 3 have been exceed 60 carriers. Author should be excluded overlapping carriers.
Response:
We included 60 patients with a pathogenic variant in CDH1 and 2 patients with a pathogenic variant in CTNNA1 in our final analysis. We hope this clarifies the question.
- Author performed sequencing of CDH1 and CTNNA1 genes for 210 of 230 patient. In table 2, the total number of patients of CDH1/CTNNA1 pathogenic variant and no pathogenic variant was 230 patients. Author should be excluded 20 patients that declined genetic testing or died before testing was conducted.
Response:
We thank the reviewer for this important point. We have excluded this patient group.
- Although author diagnosed helicobacter pylori infection for 30 patients, helicobacter pylori infection should diagnose for all gastric cancer patients.
Response:
This issue is now explained in the resuls section (lines 194-204). Helicobacter pylori infection was histopathologically diagnosed by either gastric biopsies or gastrectomy specimen in 30 patients.
- Author identified some splice site variants in this study. Author should confirm by RT-PCR and sequencing analysis whether these variants cause splicing abnormalities.
Response:
For splice variants at the conserved splice site positions +/- 1 and 2 no cDNA analysis has been performed, because variants at these positions are generally considered to affect splicing. This is also in line with the predictions of several splice predictions programs using the software Alamut (SpliceSiteFinder-like, MaxEntScan, NNSPLICE and GeneSplicer). Moreover, there are some exonic variants, which are described as splice variants (Table 4). For those variants cDNA analyses were performed previously (Lee, K. et al. Specifications of the ACMG/AMP variant curation guidelines for the analysis of germline CDH1 sequence variants. Hum. Mutat. 2018; Frebourg, T. et al. Cleft lip/palate and CDH1/E-cadherin mutations in families with hereditary diffuse gastric cancer. J. Med. Genet. 2006; Yelskaya, Z. et al. CDH1 missense variant c.1679C>G (p. T560R) completely disrupts normal splicing through creation of a novel 5′ splice site. PLoS One 2016; Vécsey-Semjén, B. et al. Novel colon cancer cell lines leading to better understanding of the diversity of respective primary cancers. Oncogene 2002)
- Page 9, line 287-290 and 325-328, these sentence and figure 2 should described in the results.
Response:
We thank the reviewer for this comment and added figure 2 and the aforementioned lines to the results.
Reviewer 5 Report
The study aims to compare disease outcome of carriers of CDH1 or CTNNA1 pathogenic variants with patients for whom no genetic cause has been identified. Further, the authors attempt to understand whether HDGC clinical testing criteria is sufficient to identify patients carrying pathogenic germline variants.
The study is potentially very interesting and it has a significant cohort. However, several issues must be addressed:
- The article would greatly benefit from English editing.
- The aim of the study is confusing and should be clearly stated.
- Line73-74 of the Introduction – The authors should re-write the sentence stating there are no reports of increased occurrence of other cancer types in CDH1 PV carriers. This is not correct. Although evidence available is not sufficient to support recommendation of additional colorectal cancer screening in CDH1 PV carriers, colorectal cancer has been reported to occur in CDH1 carriers.
- There are inconsistencies regarding the cohort number. The study starts with 230 patients, but in fact only 210 were tested. Since the study is based on the identification or not of germline variants through sequencing, this should be corrected throughout the manuscript, including in Figure 1.
- The study identified 62 patients with PV, 8 patients with benign or likely benign variants and nine patients with a variant of uncertain significance (VUS). VUS are a major challenge in HDGC clinical management, however the authors do not discuss this issue at all, which would improve the manuscript. Further, the authors should classify these patients into a separate group. For instance, it would be very interesting to see whether there is a correlation between these VUS carriers and DGC prevalence and/or occurrence of SRCC.
- Lines 157-159. This sentence should be rephrased. In fact, this is a clear drawback from the study. History of cleft lip or cleft palate should be addressed within the context of genetic counselling of HDGC families.
- Discussion is very confusing at times, with authors contradicting statements when attempting to discuss them. For instance, the paragraph encompassing lines 287-304 is one such example. The authors begin to state that GC prevalence is similar in patients with and without PV. In the following sentence, the authors attempt to explain differences in prevalence.
- The discussion around variability of expression concerning variants should also be improved. In fact, the sentence “…As already described in other…may influence penetrance.” contradicts the findings that the authors were discussing, namely the fact that the same variant yields different phenotypes in different families.
- Overall the discussion is very descriptive and lacks critical discussion of the results, namely concerning HDGC testing criteria with respect to their results.
Author Response
The study aims to compare disease outcome of carriers of CDH1 or CTNNA1 pathogenic variants with patients for whom no genetic cause has been identified. Further, the authors attempt to understand whether HDGC clinical testing criteria is sufficient to identify patients carrying pathogenic germline variants.
The study is potentially very interesting and it has a significant cohort. However, several issues must be addressed:
- The article would greatly benefit from English editing.
Response:
We thank the author for this important criticism and have updated the manuscript.
- The aim of the study is confusing and should be clearly stated.
Response:
We have addressed this issue by stating main aims of the study at the end of the introduction (lines 107-110).
- Line73-74 of the Introduction – The authors should re-write the sentence stating there are no reports of increased occurrence of other cancer types in CDH1 PV carriers. This is not correct. Although evidence available is not sufficient to support recommendation of additional colorectal cancer screening in CDH1 PV carriers, colorectal cancer has been reported to occur in CDH1 carriers.
Response:
This commentary addresses the important question of possible further cancer risks in this high-risk group. We are also aware of the described cases of colorectal cancer in patients with a CDH1 mutation. For example, our cohort included a 30-year-old patient with a colorectal carcinoma (MSS). However, the recently published guideline shows that there is no increased risk of further cancer ("Finally, no strong evidence exists to suggest that the risk of others cancer types is significantly increased in individuals with a CDH1 pathogenic variant. In particular, there is insufficient evidence to recommend additional colorectal cancer screening beyond adherence to national population screening guidelines.“ Blair et al., Lancet Oncology 2020)
- There are inconsistencies regarding the cohort number. The study starts with 230 patients, but in fact only 210 were tested. Since the study is based on the identification or not of germline variants through sequencing, this should be corrected throughout the manuscript, including in Figure 1.
Response:
We completely agree with the reviewer and have excluded this patient group. The manuscript and figure 1 was revised accordingly.
- The study identified 62 patients with PV, 8 patients with benign or likely benign variants and nine patients with a variant of uncertain significance (VUS). VUS are a major challenge in HDGC clinical management, however the authors do not discuss this issue at all, which would improve the manuscript.
Response:
We share the reviewer's view that the clinical management of patients with VUS in CDH1/CTNNA1 is very difficult and challenging. We recommend that patients with VUS have an annual gastroscopy with 30 biopsies, especially if there is a strong family history of gastric cancer or personal cancer history. We have addressed this issue in the discussions section (lines 406-410).
Further, the authors should classify these patients into a separate group. For instance, it would be very interesting to see whether there is a correlation between these VUS carriers and DGC prevalence and/or occurrence of SRCC.
Response:
We thank the reviewer for this comment and combined patients with a VUS into a separate group (results lines 312-315).
- Lines 157-159. This sentence should be rephrased. In fact, this is a clear drawback from the study. History of cleft lip or cleft palate should be addressed within the context of genetic counselling of HDGC families.
Response
We agree with the reviewer that history of cleft lip or cleft palate has to be adressed in CDH1 patients. It is part of our genetic backround workup. Only one minor child of a CDH1 PV patient has a cleft lip. So far, no genetic testing has been done in this child. We have updated this section (lines 161-163).
- Discussion is very confusing at times, with authors contradicting statements when attempting to discuss them. For instance, the paragraph encompassing lines 287-304 is one such example. The authors begin to state that GC prevalence is similar in patients with and without PV. In the following sentence, the authors attempt to explain differences in prevalence.
Response:
We agree with the reviewer that this was misleading. We have updated this part and have revised the discussion (lines 367-374).
- The discussion around variability of expression concerning variants should also be improved. In fact, the sentence “…As already described in other…may influence penetrance.” contradicts the findings that the authors were discussing, namely the fact that the same variant yields different phenotypes in different families.
Response:
We have updated this section. It is one of the most challenging parts of hereditary cancer syndromes, that variability of expression remains mysterious. This important question has to be approached in large-scale trials. Potentiel exogenous factors or polygenic causes need to be investigated. This has been edited in lines 423-426.
- Overall the discussion is very descriptive and lacks critical discussion of the results, namely concerning HDGC testing criteria with respect to their results.
Response:
We have revised this section (lines 321-340).
Round 2
Reviewer 2 Report
I consider that your changes have improved the manuscript, and that it is a very interesting work. I only request 1 clarification from the authors:
When genetic counseling is considered, it is advisable that the index case be a subject who has developed the disease.
To elucidate the equal risk of GC in both groups, the authors included an analysis of the prevalence of GC in index patients. The authors considered that "The index case in each family is the case of the original patient who met the criteria for further testing for CDH1 / CTNNA1." How many of them were subjects with CG and how many without CG in each group, at the time of being chosen as index cases?
Some minor comments:
- Page 6, line 179:“Most index patients…” instead of “ Most index, patients…”
- Page 8, line 252: …”in terms of gender preference” could be omitted
- Page 9, line 259: “These advanced….” instead of “Theses advanced..:”
- Page 10, line 338: “However, patients were not initially….” instead of … were initially”
Reviewer 3 Report
The authors did a good work and have made an effort to answer the criticisms and the article has improved. Still, there are several points that should be addressed before manuscript publication:
Main comments:
- Scientifically, it is not acceptable to present p values from statistical analysis without significant digits. Please substitute p=0.000 for a valid value along the manuscript.
- In the introduction section, the authors refer that 65% of carriers of a CDH1 pathogenic variant did not meet the 2015 HDGC clinical criteria. However, this should be interpreted with caution as the context of multiplex gene panel testing is not taken into account. Further, variant classification considers, as main criteria, variant frequency in healthy control population based on genomic databases that can be poorly curated and have limitations including low-quality data and lack of details on associated phenotypes. This may underestimate a large number of variants that can be clinically relevant. The authors should clarify their statement.
- In the results section, please specify the CDH1 pathogenic variant found in the patient with MSS colorectal cancer and the variant found in the family with midline facial defects.
- In table 2, percentages regarding cancer staging in gastrectomy should be clarified. In the group of patients without pathogenic variants, percentages were calculated taking into account the total number of gastrectomies. In contrast, percentages displayed in CDH1/CTNNA1 PV group were calculated considering only prophylactic total gastrectomies, in which diffuse gastric cancer was found (n=31). This should be corrected and results should be consistent in the corresponding results section (2.4.1 Gastric cancer) and in the discussion. The same should be elucidated regarding the percentage of patients with multifocal GC in gastrectomy.
- The 2.3 section (Helicobacter pylori infection) describes that 9% of patients without a CDH1/CTNNA1 PV were diagnosed with H. pylori. However, 18/140 corresponds to 13% - this should be corrected.
- The authors found a difference in the mean age of GC diagnosis between H. pylori infected patients that were carriers of a CDH1/CTNNA1 and those without pathogenic variants (42 ± 21 versus 52 ± 13). Nevertheless, the authors have poorly discussed this result, which suggest that H. pylori could potentiate/accelerate carcinogenesis in a CDH1 defective context.
- In the section 2.4 (Patients with CDH1 or CTNNA1 PV), the authors state that 22 different CDH1 pathogenic variants were detected in 28 families. Still, the table 4 presents 21 distinct CDH1 pathogenic variants. This should also be verified in the abstract.
- In table 4, data regarding variants of uncertain significance are very valuable. It was a bit surprising to verify that the number of family members included and the number of gastric cancer cases in each family pedigree was exactly the same. However, the authors state that from the 5 VUS carriers, only 2 were diagnosed with diffuse gastric cancer. Is it correct?
- Line 179: “Most index, patients”, remove comma.
- Line 101: “DGC” instead “diffuse gastric cancer”.
- Line 371: Remove “if a PV was detected”. It is in the sentence beginning.
- Line 449: Please correct IGCLC.
- Lines 430-433: Remove “cannot be explained”. I would suggest: “…the manifestation for the same PV of the CDH1 gene suggest that exogenous factors (smoking, alcohol abuse), endogenous factors (microbiome, hormones) or polygenic causes could be involved in the aetiology of the different clinical presentations”.
- The text needs editing and punctuation.
Reviewer 4 Report
This manuscript describes germline mutation of CDH1 and CTNNA1 genes in hereditary diffuse cancer in German population. The manuscript is revised adequately.
Reviewer 5 Report
Manuscript ID: cancers-967644 - Review Report following revision
Overall, the manuscript is improved and the authors attempted to address most of the questions raised.
However, in my opinion, the article remains highly descriptive and still lacks critical discussion of the results. For instance, the authors included data regarding the VUS cases, as well as a sentence stating that it is a challenging theme, but failed to discuss the issue and their results.
With respect to English language parameters, it still requires English editing.